# Gastric Cancer Detection with Ensemble Learning on Digital Pathology: Use Case of Gastric Cancer on GasHisSDB Dataset

**DOI:** 10.3390/diagnostics14161746

**Published:** 2024-08-12

**Authors:** Govind Rajesh Mudavadkar, Mo Deng, Salah Mohammed Awad Al-Heejawi, Isha Hemant Arora, Anne Breggia, Bilal Ahmad, Robert Christman, Stephen T. Ryan, Saeed Amal

**Affiliations:** 1College of Engineering, Northeastern University, Boston, MA 02115, USA; mudavadkar.g@northeastern.edu (G.R.M.); m.deng@northeastern.edu (M.D.); s.al-heejawi@northeastern.edu (S.M.A.A.-H.); 2Khoury College of Computer Sciences, Northeastern University, Boston, MA 02115, USA; arora.isha@northeastern.edu; 3MaineHealth Institute for Research, Scarborough, ME 04074, USA; 4Maine Medical Center, Portland, ME 04102, USA; bilal.ahmad@spectrumhcp.com (B.A.); robert.christman@spectrumhcp.com (R.C.); stephen.ryan@mainehealth.org (S.T.R.); 5The Roux Institute, Department of Bioengineering, College of Engineering at Northeastern University, Boston, MA 02115, USA

**Keywords:** cancer detection, machine learning, gastrointestinal cancer, deep learning, histopathology

## Abstract

**Simple Summary:**

Gastric cancer is a major worldwide health concern, underscoring the importance of early detection to enhance patient outcomes. Traditional histological analysis, while considered the gold standard, is labour intensive and manual. Deep learning (DL) is a potential approach, but existing models fail to extract all of the visual data required for successful categorization. This work overcomes these constraints by using ensemble models that mix different deep-learning architectures to improve classification performance for stomach cancer diagnosis. Using the Gastric Histopathology Sub-Size Images Database, the ensemble models obtained an average accuracy of more than 99% at various resolutions. ResNet50, VGGNet, and ResNet34 performed better than EfficientNet and VitNet, with the ensemble model continuously delivering higher accuracy. These findings show that ensemble models may accurately detect important characteristics from smaller picture patches, allowing pathologists to diagnose stomach cancer early and increasing patient survival rates.

**Abstract:**

Gastric cancer has become a serious worldwide health concern, emphasizing the crucial importance of early diagnosis measures to improve patient outcomes. While traditional histological image analysis is regarded as the clinical gold standard, it is labour intensive and manual. In recognition of this problem, there has been a rise in interest in the use of computer-aided diagnostic tools to help pathologists with their diagnostic efforts. In particular, deep learning (DL) has emerged as a promising solution in this sector. However, current DL models are still restricted in their ability to extract extensive visual characteristics for correct categorization. To address this limitation, this study proposes the use of ensemble models, which incorporate the capabilities of several deep-learning architectures and use aggregate knowledge of many models to improve classification performance, allowing for more accurate and efficient gastric cancer detection. To determine how well these proposed models performed, this study compared them with other works, all of which were based on the Gastric Histopathology Sub-Size Images Database, a publicly available dataset for gastric cancer. This research demonstrates that the ensemble models achieved a high detection accuracy across all sub-databases, with an average accuracy exceeding 99%. Specifically, ResNet50, VGGNet, and ResNet34 performed better than EfficientNet and VitNet. For the 80 × 80-pixel sub-database, ResNet34 exhibited an accuracy of approximately 93%, VGGNet achieved 94%, and the ensemble model excelled with 99%. In the 120 × 120-pixel sub-database, the ensemble model showed 99% accuracy, VGGNet 97%, and ResNet50 approximately 97%. For the 160 × 160-pixel sub-database, the ensemble model again achieved 99% accuracy, VGGNet 98%, ResNet50 98%, and EfficientNet 92%, highlighting the ensemble model’s superior performance across all resolutions. Overall, the ensemble model consistently provided an accuracy of 99% across the three sub-pixel categories. These findings show that ensemble models may successfully detect critical characteristics from smaller patches and achieve high performance. The findings will help pathologists diagnose gastric cancer using histopathological images, leading to earlier identification and higher patient survival rates.

## 1. Introduction

The GI tract, spanning 25 feet from the oral cavity to the anus, transports ingested substances. The digestive process begins with the oesophagus and continues through the stomach and small intestines, extracting important nutrients. Waste is then eliminated through the colon and rectum [1,2]. Tumours in these organs often result from aberrant cell growth caused by DNA changes [3]. Mutations can be caused by a variety of reasons, including health conditions, genetics, or lifestyle. The uncontrolled proliferation of malignant cells in the gastrointestinal system is caused by genetic, environmental, and lifestyle factors that interact. Common gastrointestinal (GI) cancers include oesophageal cancer, colorectal cancer, gastric cancer, bile duct cancer, anal cancer, colon cancer, gallbladder cancer, pancreatic cancer, gastrointestinal stromal tumours, liver cancer, rectal cancer, gastric cancer, and small intestine cancer. In 2020, gastric cancer was one of the top three most common cancers in 19 nations, with around 1.1 million cases reported (720,000 men, 370,000 females). Early identification of GI cancer aids in cancer treatment and reduces health-related complications.

Traditional approaches to the identification of cancer include the estimation of body fat percentage and its subsequent correlation with cancer. Other methods include identifying common microbes associated with cancer in food. Another way is the use of Indocyanine Green (ICG) in gastrointestinal surgery, which is gaining popularity, particularly for lymph node diagnosis and operative field imaging [4,5].

However, pathologists must physically assess tissue samples, which is a tough, time-consuming, and subjective procedure. Moreover, different pathologists may provide different results, making the analysis susceptible to errors. The accuracy of histopathological analysis is heavily dependent on the pathologists’ experience and knowledge, making the manual process susceptible to mistakes such as incorrect detection and diagnosis. Furthermore, a scarcity of pathologists creates significant delays in examining patient cases, potentially leading to late cancer discovery [6,7].

Various computer-aided detection (CAD) strategies have been investigated for the diagnosis of gastric cancer utilizing histopathological imaging. For more than 30 years, researchers have studied computer-aided diagnosis in gastroenterology, creating datasets from endoscopic images using various methodologies. The most widely researched issue is the identification of aberrant pathological signs in a specific location of the GI tract, notably polyps. There has also been research on the detection and categorization of disorders throughout the GI system, which has included clinical findings, anatomical markers, and therapies [8]. Figure 1 shows the steps of extraction of the histopathological image from slides.

## 2. Materials and Methods

### 2.1. Literature Reviews

Various computer-aided detection (CAD) strategies have been investigated for the diagnosis of gastric cancer utilizing histopathological imaging. For more than 30 years, researchers have studied computer-aided diagnosis in gastroenterology, creating datasets from endoscopic images using various methodologies. The most widely researched issue is the identification of aberrant pathological signs in a specific location of the GI tract, notably polyps. There has also been research on the detection and categorization of disorders throughout the GI system, which has included clinical findings, anatomical markers, and therapies [8]. During the first 20 years of development, image processing required the extraction of features using various approaches before their categorization using statistical methods [9]. These characteristics are divided into three categories: spatial, frequency, and high-level descriptors. Spatial characteristics are retrieved using pixel-based and histogram techniques, whilst frequency information is collected using Fourier and wavelet transform algorithms. High-level characteristics are retrieved with edge and region-based methods. Statistical machine learning approaches were frequently employed to categorize these characteristics.

Later, machine learning (ML) was to become commonly used in CAD to diagnose gastric cancer by extracting handmade elements such as colour, texture, and form. For this purpose, support vector machines (SVM), random forests, and Adaboost are among the most frequently employed machine learning classifiers. In recent research, deep learning has been utilized to automate the feature selection process. Several studies have shown that deep convolutional neural networks (CNN) excel at tasks such as recognizing and segmenting histopathological images related to cancer, metastasis, and genetic mutation analysis. Some investigations have indicated that these networks perform similarly to human pathologists [10,11].

Deep learning techniques have advanced significantly over the past decade, notably with the CNN architecture. This design allows for the extraction and categorization of spatial as well as high-level features, making it a major area of study for academia. Several solutions have been proposed, including hybrid approaches based on CNN attributes, transfer learning, the development of novel CNN models, and research into other deep learning networks [12].

Gastroenterology CAD research has a long history and covers a wide range of topics. For studies involving the classification of the KvasirV2 and HayperKvasir datasets, which were employed in the studies conducted [9,12], Melaku et al. (2019) utilized VGGNet and InceptionV3 to classify the Hyper KVASIR dataset with 98% accuracy using SVM. M Hmoud et al. (2020) evaluated GoogLeNet, ResNet-50, and AlexNet on the KVASIR dataset, with AlexNet outperforming the others and achieving 97% accuracy. Yogapriya et al. (2021) used VGG16, ResNet-18, and GoogLeNet on the KVASIR v2 dataset, with VGG16 leading the way with an accuracy of about 96.33%. Furthermore, Zenebe et al. (2022) proposed a unique deep convolutional neural network (CNN) with a spatial attention mechanism for categorizing gastrointestinal (GI) illnesses [13,14]. When evaluated on a dataset of 12,147 GI images, the model demonstrated an impressive accuracy of 92.84%. This study emphasizes the importance of using pre-trained models in the correct diagnosis of gastrointestinal disorders, showcasing numerous methodologies, and achieving significant advances in this sector.

The study by Weiming Hua, Chen Lia, et al., [15] centres on enhancing deep learning models for cancer detection. This research utilizes various deep learning architectures and training methodologies to boost model accuracy, achieving notable results in cancer detection. Ensemble learning, which integrates multiple models to enhance performance, has seen increasing popularity in the field of medical imaging. Previous studies, including those by Hua et al. and others, have demonstrated that ensemble methods often surpass the performance of individual models. However, the specific combination of models and their application to particular datasets, like gastric histopathology images, have not been extensively investigated. Furthermore, the referenced paper does not deeply address data preprocessing techniques, cross-validation methodologies, the selection of the best epochs, or detailed performance metrics. These areas present opportunities for further exploration to enhance the robustness and generalizability of deep learning models in medical image analysis.

Several studies on medical image analysis have investigated the possibility of deep learning models for reliable diagnosis. Varun Sapra, Luxmi Sapra, and Akashdeep Bhardwaj, et al. [16] have proposed strategies that include pruning, quantization, and knowledge distillation to minimize ensemble model size and computing costs while maintaining accuracy. While their research produced important theoretical insights, it required a thorough technical review of real datasets.

To assess how well the proposed models worked, this research evaluated them using the recently available Gastric Histopathology Sub-Size Image Database (GasHisSDB) [17]. The main contributions of this study are the development of effective deep ensemble learning models for detecting gastric cancer that outperform current other models included in the research on the GasHisSDB dataset and the development of their ability to successfully identify gastric histology images with lower resolution, perhaps resulting in a reduction of the digital scanners, data storage, and computer servers needed for histopathology activities. This may increase the chance of the early detection of gastric cancer and enhance the rates of patient survival [18,19].

### 2.2. Dataset Description

The Gastric Histopathology Sub-Size Image Database (GasHisSDB), with a link to the data available in the Appendix A, consists of 600 pictures of stomach cancer pathology obtained from a patient’s pathological slides (see Figure 1) obtained from a specific section of their gastrointestinal tract, each measuring 2048 × 2048 pixels. The images were obtained by scanning with a new USB camera at a magnification of 20. Four pathologists from Longhua Hospital at the Shanghai University of Traditional Chinese Medicine then provided tissue-specific labelling. In conjunction with five Northeastern University biomedical experts, the photos were cropped into 245,196 sub-sized gastric cancer pathology images. Two qualified pathologists from Liaoning Cancer Hospital and Institute calibrated these pictures. The dataset was then classed as aberrant or normal, with photographs reduced to three different sizes (160 × 160, 120 × 120, and 80 × 80) for each group. The dataset of gastrointestinal images is separated into three sizes: 80 × 80, 120 × 120, and 160 × 160. Each size category is further classified into two groups: abnormal and normal (see Figure 2b–d). The 80 × 80 size category has 81,390 photos, with 34,350 classed as abnormal and 47,040 labelled as normal. With regard to the 120 × 120 size category, there are 65,260 photos altogether, with 24,800 rated as abnormal and 40,460 as normal. Finally, in the 160 × 160 size group, there are 33,272 photos, with 13,117 classified as abnormal and 20155 as normal (as described in see Figure 3). In this project, the dataset of each class was divided into a five-fold cross validation.
Figure 1Process of extraction of the histopathology image.
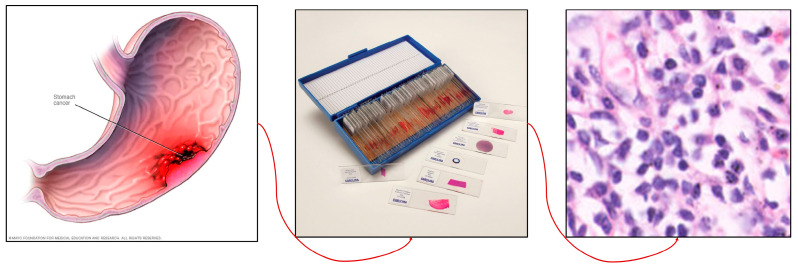



### 2.3. Methodology Overview

In this work, CNN architectures utilizing transfer learning and ensemble approaches were demonstrated to detect gastric cancer patches. The process is divided into four major steps: (1) creating the dataset by eliminating empty patches (see Figure 2a) and augmenting, (2) tailoring pre-trained networks or base models, (3) choosing the most effective base models to form ensemble models, and (4) assessing and presenting the models using different metrics and the class activation map.

To enhance the model performance, data preprocessing was conducted to create a more balanced dataset by removing non-informative empty patches. This was undertaken because the presence of these empty patches would bias the training process and therefore jeopardize the model performance. After empty patches are removed, data augmentation was employed to increase the dataset size for training.

### 2.4. Empty Patch Removal Process

Empty patches are defined as those where more than half of the pixels have an RGB intensity value greater than 230 across all channels. The following is an overview of the empty patch removal procedure, including the proportion of empty patches deleted. In the 120 × 120 resolution dataset, 15.92% of patches were removed from the abnormal subclass, while 45.01% were removed from the normal subclass. For the 160 × 160 resolution dataset, 14.77% of patches were removed from the abnormal subclass, and 44.23% were removed from the normal subclass. In the 80 × 80 resolution dataset, 17.45% of patches were removed from the abnormal subclass and 45.54% were removed from the normal subclass. After eliminating the empty patches, the remaining patches were utilized to train the model. This preprocessing phase guaranteed that the dataset contained only relevant images, allowing the model to extract detailed visual characteristics for precise categorization. Figure 4 illustrates the count differences between empty patches, which are mostly backgrounds, and non-empty patches, which represent tissues in the histopathological images.

### 2.5. Pretrained Networks as Base Models

Convolutional neural networks (CNNs) have played an important role in numerous applications since deep learning’s inception, this is due to continual advances in strength, efficiency, and adaptability. CNNs are an excellent illustration of this breakthrough, as they are particularly built for computer vision problems and use convolutional layers inspired by natural visual processes. Multiple CNN structures have evolved throughout time, each improving accuracy, speed, and overall performance, and are usually compared with the ImageNet project, a massive visual database that promotes advancements in computer vision. Historically, training CNNs from scratch required significant computer resources and time. Transfer learning (TL) provides a practical shortcut by exploiting prior information from trained models to accelerate optimization and perhaps improve classification accuracy. TL entails the transfer of weights from pre-trained models, using insights acquired from varied datasets, and increasing the speed of training processes to improve model accuracy, particularly in complicated architectures [20,21].

#### 2.5.1. ResNet34 Architecture

ResNet34 is a member of the residual networks (ResNet) family, which was introduced by He et al. in 2015. ResNet34 uses residual learning to address the issue of vanishing gradients, which is common in deep neural networks. This design is made up of 34 layers, with shortcut connections that allow gradients to flow straight across the network. These residual connections enable the training of very deep networks by overcoming the degradation issue. ResNet34 strikes a balance between depth and computational efficiency, beating shallower networks while maintaining manageable computational costs.

ResNet34 is a variation of ResNet, a CNN architecture created by Microsoft Research. ResNet34 includes 34 layers and uses residual connections to overcome the vanishing gradient problem, making training more efficient. ResNet models are popular due to their efficacy in a variety of applications.

The design starts with an input layer that processes a two-dimensional picture, followed by a 7 × 7 convolutional layer with 64 filters and a stride of 2, which includes padding to retain spatial dimensions (see Figure 5). A max pooling layer with a stride of two decreases the spatial dimensions. The network is divided into multiple stages of residual blocks. The first stage consists of three residual blocks, each with two 3 × 3 convolutional layers with 64 filters and shortcut connections. The second stage consists of four residual blocks, each with two 3 × 3 convolutional layers and 128 filters. The first block down-samples the feature maps with a stride of two, and the shortcut connection adjusts dimensions. The third stage consists of six residual blocks, each with two 3 × 3 convolutional layers and 256 filters. The first block downscales with a stride of 2 and the shortcut connection adjusts appropriately. The last stage is made up of three residual blocks, each with two 3 × 3 convolutional layers and 512 filters. The first block down-samples with a stride of two, and the shortcut connection does the same. An average pooling layer then reduces spatial dimensions to 1 × 1, and the pooled feature maps are fed via a fully connected layer with 1000 output units, which correspond with the number of classes in the ImageNet dataset.

#### 2.5.2. ResNet50 Architecture

ResNet50 is a 50-layer version of the ResNet architecture, deeper than ResNet34. This depth can improve performance on certain tasks, but it also necessitates more computational resources for training.

ResNet50’s architecture handles two-dimensional input data using a succession of layers geared for effective feature extraction. It starts with zero-padding, then adds a convolutional layer, batch normalization, ReLU activation, and max-pooling to minimize spatial dimensions (see Figure 6). The network is organized into five phases, each of which contains convolutional and identity blocks. Stage 1 prepares the data, whereas Stages 2–5 contain convolutional blocks for down-sampling and identity blocks for deep feature extraction. Following these steps, an average pooling layer reduces the spatial dimensions to one by one, followed by a flattening layer and a fully linked layer for classification. The inclusion of identity blocks allows for the direct addition of input to output, which alleviates the vanishing gradient problem and enables deeper network training. This design facilitates the effective learning of complicated characteristics, hence improving image recognition performance.

#### 2.5.3. VGGNet16 Architecture

The Visual Geometry Group (VGG) at the University of Oxford developed VGGNet16, a deep CNN architecture. It is well known for being both simple and successful in picture classification applications. VGGNet16 is made up of 16 layers, the first 13 of which are convolutional, followed by three fully connected layers. Each convolutional layer has a 3 × 3 kernel, and max-pooling layers are used after a sequence of convolutional layers to minimize spatial dimensions. Although VGGNet16 has a large number of parameters, which makes it computationally costly, it achieves good accuracy on benchmark datasets thanks to its deep architecture and consistent layer design.

VGGNet, created by the Visual Geometry Group at the University of Oxford, is renowned for its simplicity. It is mostly composed of layered convolutional layers with 3 × 3 kernels and max-pooling layers. VGGNet includes several variations, including VGG16 and VGG19, which differ in the number of layers.

The VGGNet16 architecture (see Figure 7) runs an input picture through a sequence of convolutional layers, each followed by ReLU activation. It starts with two convolutional layers of 64 filters each, followed by max-pooling (see Figure 7). This pattern is repeated with increasing numbers of filters (128, 256, and eventually 512). To minimize spatial dimensions, max-pooling is applied after each convolutional layer step. Three fully connected layers follow the convolutional layers, with the last layer serving as a softmax layer for 1000-class classification. This deep design, which makes regular use of modest 3 × 3 filters, enables good feature extraction while keeping reasonable computing cost, making VGGNet16 a strong model for image recognition applications.

#### 2.5.4. EfficientNet Architecture

EfficientNet is a class of CNN models developed by Google AI that outperforms earlier models using fewer parameters and FLOPs. They employ a novel scaling technique to improve network depth, breadth, and resolution for improved resource management.

The EfficientNet design starts with a 3 × 3 convolutional layer, then numerous mobile inverted bottleneck (MBConv) blocks with different kernel sizes and expansion factors (as shown in Figure 8). The design employs a mix of 3 × 3 and 5 × 5 MBConv blocks to efficiently record spatial characteristics at various sizes. Each MBConv block is composed of a depth wise convolution followed by a pointwise convolution, allowing the model to capture complicated patterns while being computationally efficient. The network gradually raises the number of filters and the resolution of feature maps using these MBConv blocks, resulting in a highly detailed and rich feature map. This architecture strikes a compromise between depth, breadth, and resolution, producing an efficient and strong model for picture categorization tasks.

#### 2.5.5. VitNet Architecture

The vision transformer is a novel CNN architecture created by Google Research that employs self-attention techniques seen in transformer topologies. Rather than employing convolutional layers like typical CNNs, ViT employs a transformer encoder, which allows it to recognize relationships across large distances in pictures. This method has shown outstanding results in several computer vision tasks, particularly when trained on large datasets.

The vision transformer (ViT) architecture (as shown in Figure 9) divides an input picture into fixed-size, non-overlapping patches, which are then flattened into a 1D vector (as shown in Figure 9). The vectors are then projected linearly onto a lower-dimensional space. Position embeddings are applied to these vectors in order to keep the positioning information of the patches inside the original picture. The generated sequence is routed through a transformer encoder, which consists of many layers of multi-head self-attention mechanisms and feed-forward neural networks. This encoder captures both the global environment and the interactions between patches. The output of the transformer encoder is then passed into a multi-layer perceptron (MLP) head, which computes the final predicted probability for each class. This design enables ViT to harness the benefits of transformers in collecting long-range relationships and global picture context.

#### 2.5.6. Ensemble Architecture

Convolutional neural networks (CNNs) and transfer learning have significantly increased neural network performance, but there is still potential for improvement. This article proposes the use of ensemble methods to improve the effectiveness of the three models that have been pre-trained. Ensemble learning, a machine learning and statistics-based approach combines the skills of many algorithms to extract relevant insights from data [19]. During this analysis, stacking was determined to be the most appropriate method. This requires training various ML algorithms on the information before merging them to create a composite algorithm capable of successfully combining their input.

In this research, ensemble model architecture was used using ResNet34 and VGGNet16 as basis models. Figure 10 shows the proposed ensemble, which comprises both ResNet34 and VGGNet16. Initially, each model was trained independently to determine its unique performance. We then selected the best-performing epochs for each model based on validation accuracy. The best-performing epochs were then used to build the ensemble model. To improve the robustness and generalizability of the ensemble techniques, cross-validation was used. During each fold of the cross-validation approach, the research utilized the best weights from the previous folds to train the ensemble. For example, if cross-validation was undertaken on fold 1, the best epochs from folds 2, 3, 4, and 5 of ResNet34 and VGGNet16 were used to establish the ensemble model. The ensemble model architecture was created to leverage the complementary characteristics of ResNet34 and VGGNet16. ResNet34, with its residual connections, successfully mitigates the vanishing gradient problem, allowing for deeper network training. In contrast, VGGNet16, noted for its simplicity and constant layer architecture, excels in capturing fine-grained characteristics. By integrating both models, the ensemble makes use of VGGNet16’s comprehensive feature extraction and ResNet34’s depth-wise learning capabilities. Training the ensemble model entailed freezing the early layers of both base models to preserve their pre-trained feature extraction capabilities while fine-tuning the subsequent layers to fit them to the unique dataset. This hybrid strategy enabled the ensemble model to outperform individual models, as indicated by the improved accuracy and resilience across many validation criteria. In summary, the proposed ensemble model, which includes ResNet34 and VGGNet16, showed considerable performance increases. The strategic use of cross-validation and the incorporation of complementary model architectures demonstrate the effectiveness of ensemble techniques in deep learning applications.

### 2.6. Interpretability of the Ensemble Model

Understanding how ensemble models make conclusions is critical for gaining acceptance and confidence in medical diagnoses. Our work makes use of class activation mapping (CAM) tools to display and comprehend the ensemble models’ decision-making processes. CAM identifies the key areas in histopathological pictures that have the most impact on the model’s predictions, offering insights into the characteristics examined by the models. During the training phase, we divided the dataset into a subset of validation pictures in order to determine which areas the models focused on when discriminating between normal and diseased tissues. Our cooperating pathologists then inspected the produced maps to confirm that the highlighted locations matched significant histological findings. This technique not only confirmed the model’s focus areas, but also helped to better grasp its decision-making framework.

Furthermore, we used ensemble learning with a stacking technique, in which each of the base models (ResNet34, VGGNet16, and so on) were trained separately. These models’ outputs were integrated using a meta-classifier, which pooled their predictions. This multi-model decision-making technique improves resilience and minimizes the danger of incorrect classifications by combining the capabilities of many architectures. By adding these interpretability methodologies, we may offer a clear picture of the model’s underlying workings, increasing the trustworthiness and dependability of diagnostic results.

### 2.7. Experimental Setting

The data were divided into training and validation sets. Each network was trained for 20 epochs using 5-fold cross-validation to create the model. The weights from the epoch with the best validation accuracy were chosen as the final representations for each model. Various metrics were then employed to assess accuracy, followed by many objective assessment factors to determine overall performance. The dataset was rigorously separated into training and validation sets to ensure a thorough evaluation of the model’s performance. The training approach used 5-fold cross-validation across 20 epochs for each network. This strategy provides a thorough evaluation by cycling through many train–test divides, lowering the danger of overfitting and guaranteeing that the model generalizes well to new data. For example, using an 80 × 80 dataset, we split it into five equal sections. During the initial fold, the model was trained on the first four components and verified on the fifth. In the second fold, the model was trained on parts two, three, four, and five, with validation on the first component. The third fold featured training on parts three, four, five, and one, followed by validation on part two. This method was repeated for the remaining two folds, ensuring that each portion was only used as a validation set once. After running the 20 epochs for each fold, we chose the weights from the epoch with the best validation accuracy to represent the final model for that fold. This strategy ensured that the model parameters that were picked functioned best on data that were not known, hence increasing the model’s dependability. Various metrics were then employed to assess accuracy, followed by many objective assessment factors to determine overall performance [22].

## 3. Results

The performance evaluation criteria used include accuracy, sensitivity, specificity, Jaccard index, and area under the curve (AUC). Positive samples are those that include abnormal or malignant patches, whereas negative samples contain normal or healthy patches. The phrases true positive (TP), false positive (FP), true negative (TN), and false negative (FN) are used to describe the various prediction results.

1. Accuracy: Accuracy is the ratio of properly identified samples to the total number of samples. This is computed as follows:(1)$$Accuracy=\frac{\left(TP+TN\right)}{\left(TP+TN+FP+FN\right)}$$

2. Sensitivity (recall): Sensitivity, also known as recall, is the proportion of real positive samples that the model properly identifies. This is provided as follows:(2)$$Sensitivity=\frac{\left(TP\right)}{\left(TP+FN\right)}$$

3. Specificity refers to the fraction of real negative samples properly detected by the model. This is computed as follows:(3)$$Specificity=\frac{\left(TN\right)}{\left(TN+FP\right)}$$

4. The Jaccard index, commonly known as the intersection over union (IoU), assesses the similarity between expected and observed positive samples. This is provided as follows:(4)$$Jaccard\;index=\frac{\left(TP\right)}{\left(TP+FP+FN\right)}$$

5. Area under the curve (AUC): The area under the receiver operating characteristic (ROC) curve, or AUC, relates the true positive rate (sensitivity) to the false positive rate (1-specificity). A higher AUC implies improved model performance.

By examining these measures, a thorough picture of the model’s performance can be acquired, particularly when discriminating between aberrant (positive) and normal (negative) data.

When analysing the success of machine learning models, performance measures must be considered. These metrics provide values that indicate the overall performance of a statistical or machine learning technique. In classification tasks, performance measures evaluate the model’s capacity to accurately categorize data points as well as its consistency in producing the right classifications. Classification accuracy and F1 score are both measures of classification task accuracy, whereas AUC reflects a model’s overall ability to forecast accurately. The study’s findings, which were obtained by examining these performance metrics, are shown in the Table 1, Table 2 and Table 3 below.

## 4. Discussion

In this work, the proposed model was trained on datasets with varied picture sizes, such as 80 × 80 and 120 × 120 and 160 × 160. Findings highlights the critical significance of ongoing innovation and exploration in increasing medical machine learning and hence improving healthcare practices. Notably, the findings show that the top five ensemble models had high detection accuracy across all sub-databases. The overall ensemble model demonstrated the highest accuracy, (see Table 1, Table 2 and Table 3) surpassing the performance of VGGNet, ResNet34, and ResNet50, which also outperformed VitNet and EfficientNet. The only exception was in the 160 × 160 sub-database, where EfficientNet achieved an accuracy of 92%. By combining multidisciplinary techniques and technology breakthroughs, one can pave the way towards a future in which the early and precise identification of gastrointestinal disorders is not only achievable but also common practice in protecting human health and wellbeing.

This deliberate method was used to assist the model build a solid knowledge of multiple picture dimensions, making it more flexible to a variety of real-world scenarios. Nonetheless, it is worth noting that the proposed model’s knowledge may have been expanded much more with greater computing resources. Greater computational capability might have enabled a more in-depth examination of the dataset’s intricacies, as well as the discovery of insights beyond existing capabilities.

These advanced methodologies offer various structures and improvement tactics that may increase the efficacy of the model. Time restrictions precluded the incorporation of these algorithms into the current system; nonetheless, their use holds great promise for improving cancer detection techniques.

This research prioritizes in innovation and placing people at the heart of the research. The objective is to make research applications as user-friendly as possible while also emphasizing how they might benefit medical practitioners. This research strives to integrate the most recent research work with practical applications, avoiding plagiarism and crafting a research story that is honest and true to the research’s commitment to expanding scientific understanding.

### Experimental Setting

In the future, the focus will be on two major goals: developing an easy-to-use web app for cancer detection and expanding current algorithms to include new forms of cancer. The primary goal of this research is to develop a model that will work for all forms of cancer, not just specific forms. This will be accomplished by utilizing many data sources and innovative algorithms in order to obtain a thorough understanding of cancer, hence assisting healthcare professionals with diagnosis.

By integrating multidisciplinary methodologies and capitalizing on technological developments, we hope to contribute towards a future in which the early and precise identification of GI problems is not only possible, but is also the standard for protecting human health and wellbeing.

## 5. Conclusions

Detecting and diagnosing gastrointestinal (GI) illnesses is critical for human health, yet it can be challenging owing to limited medical competence and expensive expenses. The use of machine learning, particularly deep learning techniques, has the potential to increase the speed and accuracy of GI illness identification. This research study investigated the efficiency of ensemble approaches using five pre-trained models on the Gastric Histopathology Sub-size Image Database (GasHisSDB), which comprises a diverse set of pictures in various pixel sizes and categories. Considerable boost in prediction accuracy can be seen when utilizing ensemble learning, which combines the predictive skills of several models, as opposed to using individual models and in contrast with previous studies that used comparable datasets. This demonstrates the potential of ensemble approaches to improve the capabilities of medical machine learning systems, resulting in more effective and precise diagnoses. The current approach is based on transfer learning, a technique that improves model learning by leveraging knowledge from previously trained models. Moreover, we employed an ensemble strategy to improve performance by merging various classifiers. Following a rigorous review, the current approach revealed a good accuracy for the test dataset, exceeding current evaluation techniques. This demonstrates how deep learning may help alleviate the pressure on healthcare systems while also improving human health outcomes [22,23,24,25]. In this study, advanced deep ensemble learning models were created that used transfer learning from multiple pre-trained networks, including VitNet, EfficientNet, VGGNet, ResNet34, and ResNet50, to improve stomach cancer diagnosis. The study found that using base models in ensemble learning resulted in high identification accuracy (97.57% to 99.72%) for histopathology pictures with resolutions ranging from 80 × 80 pixels to 160 × 160 pixels. The experimental results demonstrate that ensemble models may extract key information even from smaller picture patches while retaining good performance. This improvement implies the possibility of adopting digital scanners with lower specifications, as well as reduced data storage and computing needs for histopathology operations. This, in turn, might increase the speed of stomach cancer identification and perhaps increase survival rates. Continued work in these areas is intended to push the boundaries of medical image processing and enhance clinical results.

However, it is critical to recognize the limits of the current research. The usage of a restricted dataset emphasizes the need to have access to larger, higher-quality datasets to enhance and confirm the approaches. Furthermore, computational restrictions may have influenced the scope of the results. Future studies might seek to introduce new preprocessing methods and to optimize algorithms to boost performance. Furthermore, the field of medical image retrieval offers several chances for continuous research, including the ability to use multiple deep learning approaches and models for complete examination.

## Figures and Tables

**Figure 2 diagnostics-14-01746-f002:**
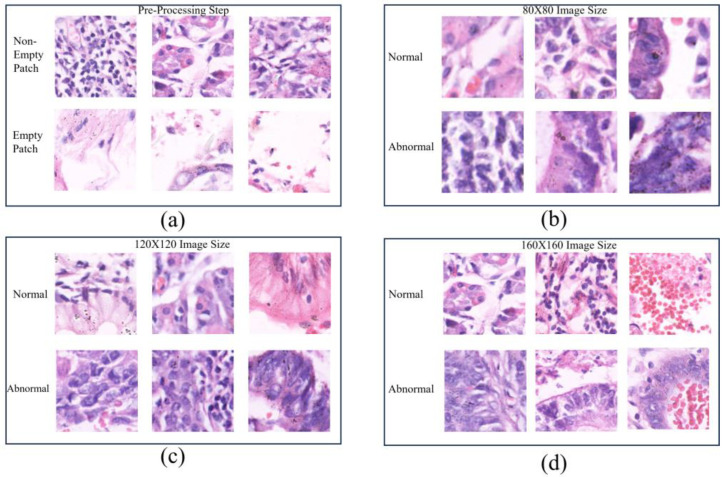
Examples of histopathological gastric images. (**a**) The pre-processing step and (**b**–**d**) examples of different image sizes.

**Figure 3 diagnostics-14-01746-f003:**
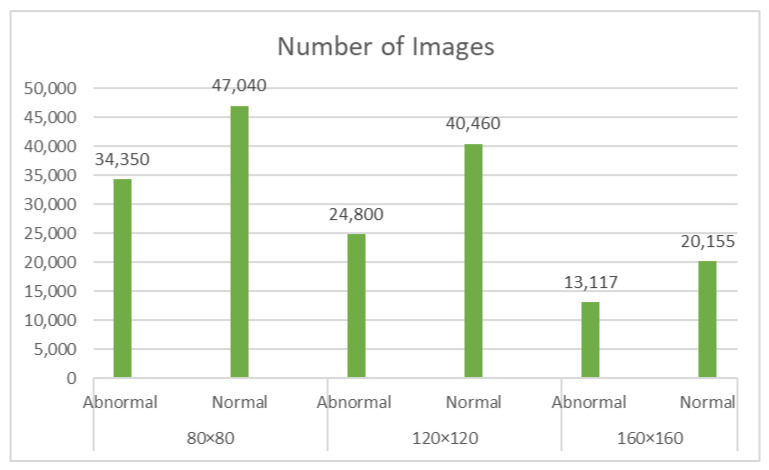
Summaries of the total number of images for every subclass in an experiment setup.

**Figure 4 diagnostics-14-01746-f004:**
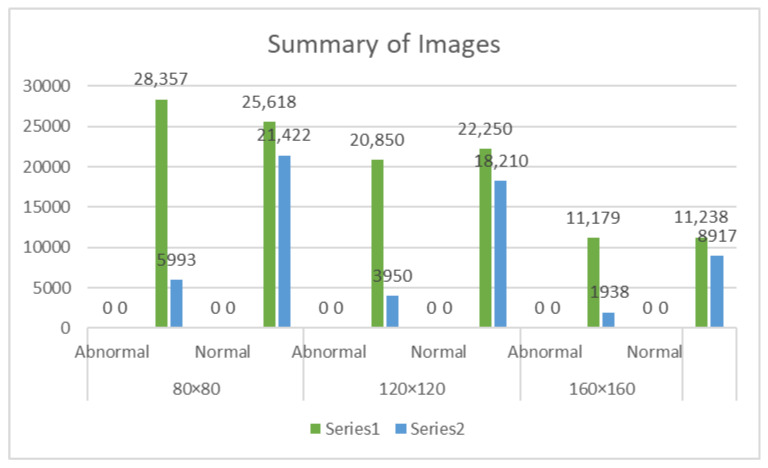
Summarises the total number of the empty and non-empty patch images in the dataset.

**Figure 5 diagnostics-14-01746-f005:**
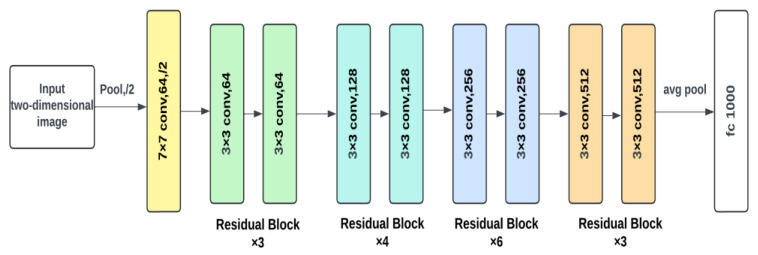
ResNet34 model architecture.

**Figure 6 diagnostics-14-01746-f006:**
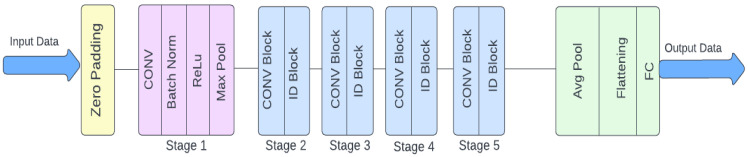
ResNet50 model architecture.

**Figure 7 diagnostics-14-01746-f007:**
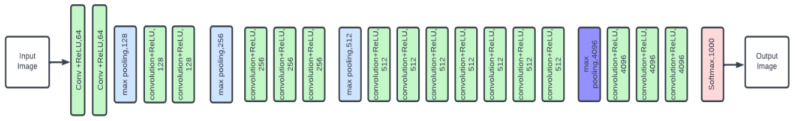
VGGNet16 model architecture.

**Figure 8 diagnostics-14-01746-f008:**
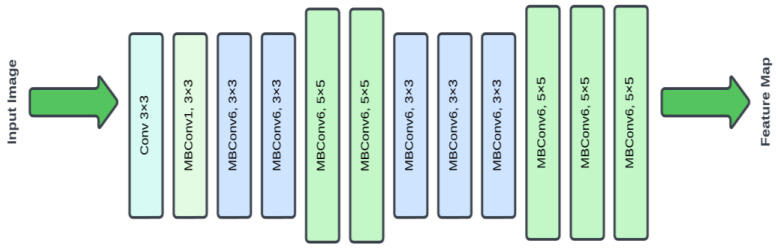
EfficientNet model architecture.

**Figure 9 diagnostics-14-01746-f009:**
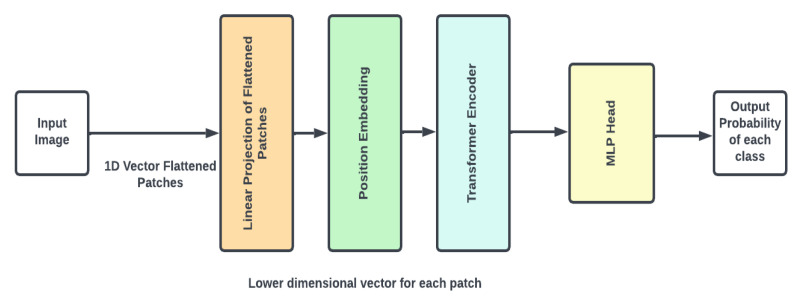
VITNet model architecture.

**Figure 10 diagnostics-14-01746-f010:**
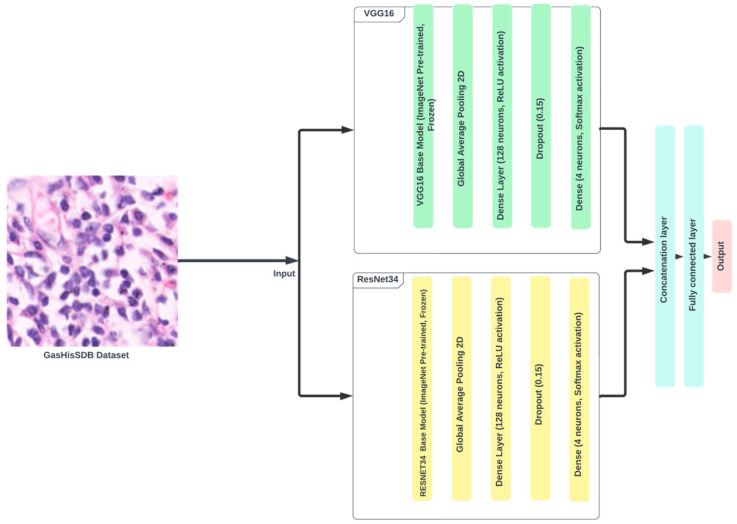
Ensemble model detailed architecture.

**Table 1 diagnostics-14-01746-t001:** The effectiveness of the several deep learning models was assessed using an 80-pixel sub-database, displayed above.

Model	Fold	Train Accuracy	Train Loss	Val Accuracy	Val Loss	Jaccard Index	AUC	Specificity	Sensitivity
**Resnet34**	1	98.7508	0.0331	93.6131	0.2175	0.7447	0.9719	0.9604	0.8494
	2	97.8192	0.0604	93.7206	0.2092	0.7472	0.9717	0.9638	0.8427
	3	98.8546	0.0329	93.8111	0.2210	0.7384	0.9760	0.9699	0.8204
	**4**	**97.7773**	**0.0596**	**93.8226**	**0.1724**	**0.7466**	**0.9777**	**0.9625**	**0.8491**
	5	98.1767	0.0487	93.7910	0.2399	0.7640	0.9754	0.9518	0.8900
**ResNet50**	1	98.4818	0.0409	93.7408	0.1936	0.8763	0.8979	0.9682	0.8277
	**2**	**96.2419**	**0.0958**	**94.2439**	**0.1880**	**0.8860**	**0.9069**	**0.9703**	**0.8435**
	3	98.4410	0.0413	93.4971	0.2024	0.8742	0.9042	0.9577	0.8508
	4	97.0654	0.0769	92.7688	0.2016	0.8601	0.9071	0.9431	0.8711
	5	93.5411	0.1673	93.8459	0.1869	0.8772	0.9064	0.9648	0.8480
**VitNet**	1	83.6243	0.3631	83.7591	0.3555	0.6945	0.7132	0.9347	0.4916
	2	82.9835	0.3816	84.2836	0.3490	0.6969	0.7076	0.9493	0.4660
	3	78.4974	0.4755	78.8656	0.4560	0.5946	0.5054	0.9988	0.0121
	**4**	**82.5612**	**0.3898**	**84.2841**	**0.3505**	**0.6986**	**0.7015**	**0.9488**	**0.4542**
	5	78.3217	0.5231	77.4105	0.5353	0.5674	0.5	1	0
**VggNet**	1	98.7371	0.0366	93.7591	0.2579	0.8763	0.9746	0.9747	0.8053
	**2**	**96.2693**	**0.0969**	**94.3341**	**0.1785**	**0.8868**	**0.9757**	**0.9738**	**0.8353**
	3	98.5683	0.0417	93.9220	0.2120	0.8763	0.9751	0.9720	0.8178
	4	94.3224	0.1519	93.7681	0.1621	0.8756	0.9733	0.9685	0.8245
	5	98.3636	0.0487	93.7910	0.2329	0.8768	0.9724	0.9693	0.8302
**EfficientNet**	1	74.9474	0.5035	75.4057	0.5142	0.6094	0.7478	0.7172	0.7784
	2	82.4435	0.3866	83.0474	0.3685	0.6829	0.6726	0.9421	0.4030
	3	81.0359	0.4139	82.3709	0.3855	0.6704	0.6346	0.9463	0.3229
	**4**	**82.3379**	**0.3879**	**84.2231**	**0.3531**	**0.6975**	**0.6725**	**0.9419**	**0.4030**
	5	82.6571	0.3805	83.5210	0.3651	0.6944	0.6816	0.9401	0.4232
**Ensemble**	**1**	**98.3587**	**0.0436**	**99.3430**	**0.0252**	**0.9867**	**0.9904**	**0.9957**	**0.9850**
	2	98.5187	0.0410	99.2421	0.0211	0.9839	0.9875	0.9962	0.9787
	3	98.6001	0.0384	99.2056	0.0237	0.9836	0.9898	0.9936	0.9861
	4	97.9234	0.0627	97.8197	0.0651	0.9562	0.9642	0.9886	0.9398
	5	98.5642	0.0390	99.0869	0.0221	0.9823	0.9866	0.9943	0.9789

The most interesting results are shown in bold.

**Table 2 diagnostics-14-01746-t002:** The effectiveness of the several deep learning models was assessed using a 120-pixel sub-database, displayed above.

Model	Fold	Train Accuracy	Train Loss	Val Accuracy	Val Loss	Jaccard Index	AUC	Specificity	Sensitivity
**Resnet34**	1	98.8627	0.0327	96.8167	0.1097	0.8347	0.9879	0.9812	0.9075
	**2**	**99.2847**	**0.0187**	**97.0297**	**0.1327**	**0.8444**	**0.9815**	**0.9884**	**0.8884**
	3	98.9145	0.0336	96.9364	0.1025	0.8392	0.9907	0.9901	0.8762
	4	99.1640	0.0234	96.4475	0.1208	0.8175	0.9879	0.9836	0.8778
	5	99.2957	0.0214	96.6893	0.1074	0.8149	0.9897	0.9877	0.8642
**ResNet50**	1	99.5552	0.0125	96.7257	0.1120	0.9316	0.9518	0.9756	0.9281
	2	99.2449	0.0213	96.9856	0.1156	0.9372	0.9254	0.9951	0.8557
	**3**	**99.7229**	**0.0089**	**97.0930**	**0.1198**	**0.9388**	**0.9493**	**0.9833**	**0.9154**
	4	99.5007	0.0142	96.8166	0.1253	0.9339	0.9379	0.9853	0.8905
	5	99.5042	0.0147	97.0748	0.0995	0.9379	0.9406	0.9860	0.8951
**VitNet**	1	86.6963	0.3109	87.4715	0.2875	0.7518	0.7395	0.9488	0.5301
	2	84.6948	0.3495	86.1826	0.3185	0.7254	0.7123	0.9470	0.4775
	3	82.2817	0.4486	81.7531	0.4287	0.6310	0.5	1	0
	**4**	**85.7447**	**0.3298**	**87.9123**	**0.2929**	**0.7581**	**0.7607**	**0.9464**	**0.5750**
	5	84.2084	0.3465	86.3265	0.3085	0.7231	0.6767	0.9582	0.3951
**VggNet**	1	76.6185	0.4764	75.6162	0.5053	0.6152	0.7657	0.7776	0.7538
	**2**	**88.3740**	**0.2758**	**89.1768**	**0.2611**	**0.7889**	**0.7476**	**0.9592**	**0.5361**
	3	88.1351	0.2822	88.3388	0.2732	0.7756	0.7372	0.9605	0.5139
	4	86.6576	0.3140	88.2826	0.2813	0.7673	0.6979	0.9591	0.4368
	5	88.3702	0.2745	88.4198	0.2781	0.7779	0.7460	0.9594	0.5325
**EfficientNet**	1	99.1949	0.0265	96.8167	0.1372	0.9340	0.9878	0.9867	0.8818
	2	99.8978	0.0054	96.9416	0.1716	0.9363	0.9886	0.9873	0.8884
	**3**	**98.4339**	**0.0464**	**97.1824**	**0.1130**	**0.9403**	**0.9883**	**0.9901**	**0.8897**
	4	96.4544	0.1011	96.0322	0.1098	0.9176	0.9860	0.9757	0.8905
	5	99.8366	0.0061	96.9614	0.1410	0.9338	0.9902	0.98363	0.9005
**Ensemble**	1	97.2581	0.0728	97.6125	0.0646	0.9501	0.9593	0.9853	0.9332
	2	97.7973	0.0622	98.1738	0.0533	0.9627	0.9647	0.9913	0.9381
	3	97.7894	0.0603	97.8756	0.0618	0.9531	0.9636	0.9874	0.9399
	4	97.8457	0.0625	97.5778	0.0685	0.9486	0.9549	0.9876	0.9223
	**5**	**98.9735**	**0.0316**	**99.4250**	**0.0226**	**0.9872**	**0.9895**	**0.9965**	**0.9826**

The most interesting results are shown in bold.

**Table 3 diagnostics-14-01746-t003:** The effectiveness of the several deep learning models was assessed using a 160-pixel sub-database, displayed above.

Model	Fold	Train Accuracy	Train Loss	Val Accuracy	Val Loss	Jaccard Index	AUC	Specificity	Sensitivity
**Resnet34**	1	99.6311	0.0132	97.9807	0.0648	0.8949	0.9961	0.9887	0.9398
	2	99.5496	0.0148	98.1776	0.0656	0.9022	0.9913	0.9922	0.9341
	3	99.7337	0.0083	98.1038	0.0535	0.8928	0.9976	0.9912	0.9308
	4	97.1184	0.0896	97.2005	0.0821	0.8665	0.9931	0.9806	0.9361
	**5**	**99.5437**	**0.0213**	**98.4187**	**0.0738**	**0.9105**	**0.9929**	**0.9862**	**0.9739**
**ResNet50**	**1**	**99.3660**	**0.0288**	**98.4396**	**0.0579**	**0.9678**	**0.9739**	**0.9904**	**0.9573**
	2	99.5842	0.0170	98.3143	0.0592	0.9628	0.9719	0.9894	0.9544
	3	99.8263	0.0066	98.1038	0.0776	0.9599	0.9684	0.9874	0.9494
	4	99.9538	0.0018	97.9348	0.0743	0.9574	0.9620	0.9903	0.9338
	5	99.7718	0.0112	98.3708	0.0625	0.9648	0.9832	0.9839	0.9826
**VitNet**	1	82.2614	0.4526	81.6888	0.4532	0.6295	0.5	1	0
	2	84.4919	0.3426	86.2870	0.3141	0.7307	0.6980	0.9555	0.4405
	3	81.9212	0.4638	83.0248	0.4527	0.6543	0.5	1	0
	4	84.3706	0.3502	84.6718	0.3393	0.6977	0.6841	0.9498	0.4184
	**5**	**82.0483**	**0.3951**	**86.3919**	**0.3069**	**0.7274**	**0.6337**	**0.9776**	**0.2898**
**VggNet**	1	77.3246	0.4784	76.5826	0.4820	0.6291	0.7732	0.7643	0.7822
	2	90.0875	0.2296	93.0702	0.1838	0.8607	0.7924	0.9604	0.6244
	**3**	**91.0623**	**0.2155**	**92.8018**	**0.1911**	**0.8572**	**0.8139**	**0.9641**	**0.6636**
	4	91.0069	0.2178	93.0925	0.1655	0.8589	0.8168	0.9628	0.6709
	5	91.4476	0.2077	92.0146	0.1945	0.8444	0.8186	0.9659	0.6712
**EfficientNet**	**1**	**99.9769**	**0.0014**	**98.2101**	**0.1121**	**0.9623**	**0.9946**	**0.9915**	**0.9398**
	2	99.6882	0.0098	97.9498	0.0795	0.9584	0.9944	0.9952	0.9088
	3	98.8773	0.0347	97.9683	0.0896	0.9579	0.9921	0.9934	0.9122
	4	99.2738	0.0244	97.2005	0.0940	0.9464	0.9912	0.9834	0.9243
	5	99.6692	0.0091	98.1312	0.0814	0.9586	0.9948	0.9896	0.9391
**Ensemble**	1	98.4785	0.0412	98.8067	0.0313	0.9751	0.9829	0.9910	0.9749
	2	99.0877	0.0268	98.9066	0.0325	0.9753	0.9775	0.9955	0.9594
	3	99.1203	0.0298	99.1873	0.0260	0.9830	0.9866	0.9945	0.9787
	**4**	**99.4813**	**0.0150**	**99.7246**	**0.0079**	**0.9942**	**0.9964**	**0.9977**	**0.9952**
	5	98.9165	0.0319	99.3291	0.0245	0.9836	0.9901	0.9948	0.9855

The most stunning results are shown in bold.

## Data Availability

The data presented in this study are available in this article.

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
