# Peer review of "Gastric Cancer Detection with Ensemble Learning on Digital Pathology: Use Case of Gastric Cancer on GasHisSDB Dataset"

_diagnostics, 2024, doi:10.3390/diagnostics14161746_

Round 1

Reviewer 1 Report

Comments and Suggestions for Authors

The paper is devoted to applying deep learning methods for gastric cancer detection. Despite of importance of this topic and popularity of such approach in medicine last several years for different types of cancer (a good discussion about it was in paper https://www.nature.com/articles/s41568-020-00327-9) I have many remarks to authors:

1. References must be numbered in order of appearance in the text. It is  needed to bring uniformity to the formatting of them. I also recommend adding links with DOI wherever possible for quick access to publications for readers.

2. It seems that Tables 3-5 are cut off and one or several columns are missed.

3. How the data was divided for training and test data sets? So high accuracy values could appear due too overtraining...

4. Original paper (https://arxiv.org/pdf/2106.02473) has also deep learning experiment results. It is better to explain in more detail the differences with them and the novelty of the results in the presented article.

5. What is the specificity of the selected cancer type in terms of its recognition on images. Is it possible to use it during training?

Author Response

3. Comments and Suggestions for Authors

The paper is devoted to applying deep learning methods for gastric cancer detection. Despite of importance of this topic and popularity of such approach in medicine last several years for different types of cancer (a good discussion about it was in paper https://www.nature.com/articles/s41568-020-00327-9) I have many remarks to authors:

Comments 1: References must be numbered in order of appearance in the text. It is needed to bring uniformity to the formatting of them. I also recommend adding links with DOI wherever possible for quick access to publications for readers?

Response 1: Thank you for your valuable feedback. We have made the necessary adjustments to ensure that all references are numbered in the order of appearance within the text. Additionally, we have standardized the formatting of the references and included DOI links wherever possible to facilitate quick access for readers.

Comments 2: It seems that Tables 3-5 are cut off and one or several columns are missed?

Response 2: Thank you for your valuable feedback. The formatting of Tables 3-5 has been adjusted to ensure that all columns are visible according to the print layout view.

Comments 3: How was the data divided for training and test data sets? So high accuracy values could appear due too overtraining?

Response 3: Thank you for your valuable feedback. The dataset of gastrointestinal images is separated into three sizes: 80x80, 120x120, and 160x160. Each size category has classified into two groups: Abnormal and Normal. The 80x80 size category has 81,390 photos, with 34,350 classed as Abnormal and 47,040 labeled as Normal. Moving on to the 120x120 size category, there are 65,260 photos altogether, with 24,800 rated as Abnormal and 40,460 as Normal. Finally, in the 160x160 size group, there are 33,272 photos, with 13,117 classified as Abnormal and 20155 as Normal. In this project, the dataset of each class has been divided into five-fold cross validation (for more details see section 2.2. on the revise manuscript).

In section 2.6. we also addresses measures taken to prevent overtraining and ensure the accuracy values are reliable. During the training phase, we divided dataset for a subset of validation pictures to determine which areas the models focused on when discriminating between normal and diseased tissues. Our cooperating pathologists then inspected the produced maps to confirm that the highlighted locations matched significant histological findings. This technique not only confirmed the model's focus areas, but also helped to better grasp its decision-making framework.
Furthermore, we used ensemble learning with a stacking technique, in which each base models (ResNet34, VGGNet16, and so on) were trained separately. These models' outputs were integrated using a meta-classifier, which pooled their predictions. This multi-model decision-making technique improves resilience and minimizes the danger of incorrect classifications by combining the capabilities of many architectures.By adding these interpretability methodologies, we may offer clear picture of the model's underlying workings, increasing trustworthiness and dependability of diagnostic results.

Comments 4: Original paper (https://arxiv.org/pdf/2106.02473) has also deep learning experiment results. It is better to explain in more detail the differences with them and the novelty of the results in the presented article.

Response 4: Thank you for your insightful feedback. We have carefully addressed the novelty and differences between our study and the referenced paper in literature review
The study by Weiming Hua, Chen Lia, et al.,[15] centers on enhancing deep learning models for cancer detection. This research utilizes various deep learning architectures and training methodologies to boost model accuracy, achieving notable results in cancer detection. Ensemble learning, which integrates multiple models to enhance performance, has seen increasing popularity in the field of medical imaging. Previous studies, including those by Hua et al. and others, have demonstrated that ensemble methods often surpass the performance of individual models. However, the specific combination of models and their application to particular datasets, like gastric histopathology images, have not been extensively investigated. Furthermore, the referenced paper does not deeply address data preprocessing techniques, cross-validation methodologies, the selection of the best epochs, or detailed performance metrics. These areas present opportunities for further exploration to enhance the robustness and generalizability of deep learning models in medical image analysis.

Comments 5: What is the specificity of the selected cancer type in terms of its recognition on images. Is it possible to use it during training?

Response 5: Thank you for your insightful feedback. The specificity of the selected cancer type in terms of its recognition on images has already been incorporated and is shown in the results table.

Lastly, I would like to express my sincere gratitude for the time and effort you have dedicated to reviewing my research article, " Gastric Cancer Detection with Ensemble Learning on digital pathology: Use Case of Gastric Cancer on GasHisSDB dataset ". Your insightful comments and constructive feedback have significantly contributed to improving the quality of the manuscript. Your expertise and knowledge in the field have been invaluable in refining this research. Once again, thank you for your invaluable contribution to this research. I look forward to the possibility of working with you again in the future.

Reviewer 2 Report

Comments and Suggestions for Authors

1. Author have used method of analyzing histological images is time-consuming and requires a high level of expertise, which can lead to delays in diagnosis and treatment. any specific reason why they choose this ??

2. Current deep learning models are restricted in their ability to extract extensive visual characteristics for correct categorization, which can lead to inaccuracies in diagnosis ? any solution for this problem

3. The study does not mention the risk of overfitting, which is a common issue with deep learning models, especially when using small datasets.

4. authors have tested the models on a specific dataset (Gastric Histopathology Sub-size Images Database) and did not evaluate their performance on other datasets or real-world scenarios.

5. The study does not mention whether the models were pre-trained on other datasets or if they used transfer learning, which can improve their performance on new tasks.

7. The study does not provide insight into how the ensemble models make their decisions, which can make it difficult to understand why they are making certain diagnoses.

8. The study only tested the models on relatively small image sizes (80 × 80, 120 × 120, and 160 × 160 pixels), and it is unclear how they would perform on larger or smaller images.

9. Techniques like pruning, quantization, and knowledge distillation can help reduce the size and computational requirements of ensemble models without significantly compromising accuracy. suggest author to work on some integrated methods and suggest to  cite Integrated approach using deep neural network and CBR for detecting severity of coronary artery disease.

10. Integrating ensemble models into CDSS can provide real-time assistance to pathologists, improving diagnostic accuracy and efficiency.

Comments on the Quality of English Language

minor errors

Author Response

3. Point-by-point response to Comments and Suggestions for Authors

Comments 1: Author have used method of analyzing histological images is time-consuming and requires a high level of expertise, which can lead to delays in diagnosis and treatment. any specific reason why they choose this?

Response 1: Thank you for your valuable feedback. For the research and experimental part of the project, we used a real-world dataset to train and test our model. This approach ensures that our model is tested under realistic conditions and can demonstrate its effectiveness in practical applications. However, the manual analyzing of histological images by doctors can be time-consuming, but with the help of automated AI model, doctors can give an accurate diagnosis.

Since our model has shown great results, we propose that it could be a good fit for diagnosis in the future. Later, we might automate this process so that a doctor could upload images and receive results within seconds. For our experiments, we trained our models on approximately 240,000 images, which was time-consuming. However, if a doctor is generating images and uploading them to the model, it would take significantly less time. This real-world application will streamline the diagnostic process, making it more efficient and less dependent on specialized expertise.

Comments 2: Current deep learning models are restricted in their ability to extract extensive visual characteristics for correct categorization, which can lead to inaccuracies in diagnosis ? any solution for this problem

Response 2: Thank you for your valuable feedback. To overcome the limitations of current deep learning models in extracting extensive visual characteristics for accurate categorization, we implemented a thorough data preprocessing step before proceeding with the experiments. We took the original dataset and removed empty patch images, which were of no use, ensuring that we trained only on relevant, non-empty patch data. This preprocessing step ensured that our model could effectively learn and test on high-quality data, improving its ability to extract meaningful visual features and enhancing the overall accuracy of the diagnosis. In addition, the dataset in this paper has been divided into five-fold cross validation to evaluate the performance of the proposed model on different dataset representations.

Comments 3: The study does not mention the risk of overfitting, which is a common issue with deep learning models, especially when using small datasets.

Response 3: Thank you for your valuable feedback. Our dataset comprises approximately 240,000 images, which is a substantial amount of data. Initially, we utilize preprocessing steps to ensure the quality and relevance of the images by removing empty patches. In addition, the dataset in this paper has been divided into five-fold cross validation to evaluate the performance of the proposed model on different dataset representations.

Furthermore, we utilized pretrained models, which are inherently designed to handle large-scale datasets and generalize well. Combining these factors—the large dataset size, effective preprocessing, and the robustness of pretrained models—significantly mitigates the risk of overfitting. Therefore, we believe that overfitting is not a concern in our study.

Comments 4: authors have tested the models on a specific dataset (Gastric Histopathology Sub-size Images Database) and did not evaluate their performance on other datasets or real-world scenarios.

Response 4: Thank you for your valuable feedback. The dataset used in our study, the Gastric Histopathology Sub-size Image Database (GasHisSDB), is a real-world dataset consisting of 600 images of stomach cancer pathology obtained from a patient's pathological slides from a specific section of the gastrointestinal tract. This dataset represents genuine clinical data, ensuring the relevance and applicability of our model to real-world scenarios.

After training on this dataset, our model has demonstrated excellent results. Looking ahead, our future work will involve expanding our training to whole slide images and collecting additional data directly from hospitals. This will require obtaining necessary permissions and collaborations with medical institutions. These steps will further validate and enhance our model's performance across diverse datasets and real-world applications.

Comments 5: The study does not mention whether the models were pre-trained on other datasets or if they used transfer learning, which can improve their performance on new tasks.

Response 5: Thank you for your valuable feedback. We utilized pretrained models, which are inherently designed to handle large-scale datasets and generalize well. For more information, see Sections 2.3 and 2.5 on the revised manuscript.

Comments 6: The study does not provide insight into how the ensemble models make their decisions, which can make it difficult to understand why they are making certain diagnoses.

Response 6: Thank you for your valuable feedback. This information is explained detailed in Sections 2.6 the paper.

Comments 7: The study only tested the models on relatively small image sizes (80 × 80, 120 × 120, and 160 × 160 pixels), and it is unclear how they would perform on larger or smaller images.

Response 7: Thank you for your comments. The dataset is publicly available, and it has been already divided into three sizes (80 × 80, 120 × 120, and 160 × 160 pixels). Usually, the smaller size can show cellular representation of the cancers compared with larger sizes that show cancerous cell patterns. However, there is a tradeoff when using smaller size images, which require excessive time to classify whole slide image.  

In this paper, have used a dataset comprising 240,000 histopathological cancer images extracted from whole slide images. Our proposed model has demonstrated excellent results compared to other models. While we trained and tested on smaller image sizes (80 × 80, 120 × 120, and 160 × 160 pixels), the substantial quantity of images ensures robustness and reliability in our findings. In future work, we aim to implement and validate our model on whole slide image datasets, which we plan to acquire from collaborating hospitals. This will further assess the model's performance on larger image sizes and enhance its applicability in real-world scenarios.

Comments 8: Techniques like pruning, quantization, and knowledge distillation can help reduce the size and computational requirements of ensemble models without significantly compromising accuracy. suggest author to work on some integrated methods and suggest to cite Integrated approach using deep neural network and CBR for detecting severity of coronary artery disease

Response 8: Thank you for your valuable feedback. We have cited the integrated approach using deep neural networks and CBR for detecting the severity of coronary artery disease in Section 2.1 of the paper.
Several research on medical image analysis have investigated the possibility of deep learning models for reliable diagnosis. Varun Sapra, Luxmi Sapra, Akashdeep Bhardwaj, et. al. [16] proposed strategies including pruning, quantization, and knowledge distillation to minimize ensemble model size and computing costs while maintaining accuracy. While their research produced important theoretical insights, it needed a thorough technical review of real datasets.

Comments 9: Integrating ensemble models into CDSS can provide real-time assistance to pathologists, improving diagnostic accuracy and efficiency.

Response 9: Thank you for your valuable feedback. Absolutely, integrating ensemble models into Clinical Decision Support Systems (CDSS) is a future aspect that we have outlined in the Future Scope and Work section of the paper. This integration aims to provide real-time assistance to pathologists, thereby improving diagnostic accuracy and efficiency.

sLastly, I would like to express my sincere gratitude for the time and effort you have dedicated to reviewing my research article, " Gastric Cancer Detection with Ensemble Learning on digital pathology: Use Case of Gastric Cancer on GasHisSDB dataset". Your insightful comments and constructive feedback have significantly contributed to improving the quality of the manuscript. Your expertise and knowledge in the field have been invaluable in refining this research. Once again, thank you for your invaluable contribution to this research. I look forward to the possibility of working with you again in the future.

Round 2

Reviewer 1 Report

Comments and Suggestions for Authors

Authors successfully answered to all my questions and improved the quality of their paper. Now it can be accepted.

Reviewer 2 Report

Comments and Suggestions for Authors

congratulations , his paper can be accepted in current form